# Advanced Imaging in Multiple Myeloma: New Frontiers for MRI

**DOI:** 10.3390/diagnostics12092182

**Published:** 2022-09-09

**Authors:** Pooya Torkian, Javid Azadbakht, Pietro Andrea Bonaffini, Behrang Amini, Majid Chalian

**Affiliations:** 1Vascular and Interventional Radiology, Department of Radiology, University of Minnesota, Minneapolis, MN 55455, USA; 2Department of Radiology, Kashan University of Medical Sciences, Kashan 8715988141, Iran; 3Department of Radiology, Papa Giovanni XXIII Hospital, Piazza OMS, 1, 24127 Bergamo, Italy; 4Department of Musculoskeletal Imaging, The University of Texas MD Anderson Cancer Center, Huoston, TX 77030, USA; 5Musculoskeletal Imaging and Intervention, Department of Radiology, University of Washington, Seattle, WA 98105, USA

**Keywords:** multiple myeloma, plasma cell dyscrasia, diffusion weighted imaging, whole body MRI

## Abstract

Plasma cell dyscrasias are estimated to newly affect almost 40,000 people in 2022. They fall on a spectrum of diseases ranging from relatively benign to malignant, the malignant end of the spectrum being multiple myeloma (MM). The International Myeloma Working Group (IMWG) has traditionally outlined the diagnostic criteria and therapeutic management of MM. In the last two decades, novel imaging techniques have been employed for MM to provide more information that can guide not only diagnosis and staging, but also treatment efficacy. These imaging techniques, due to their low invasiveness and high reliability, have gained significant clinical attention and have already changed the clinical practice. The development of functional MRI sequences such as diffusion weighted imaging (DWI) or intravoxel incoherent motion (IVIM) has made the functional assessment of lesions feasible. Moreover, the growing availability of positron emission tomography (PET)–magnetic resonance imaging (MRI) scanners is leading to the potential combination of sensitive anatomical and functional information in a single step. This paper provides an organized framework for evaluating the benefits and challenges of novel and more functional imaging techniques used for the management of patients with plasma cell dyscrasias, notably MM.

## 1. Introduction

Multiple myeloma (MM), the second most prevalent hematopoietic malignancy, is a plasma cell dyscrasia where monoclonal plasma cells infiltrate bone marrow, resulting in marrow failure and/or bone destruction, mainly in the axial skeleton and proximal appendicular bones [1,2]. MM exists on the end of a spectrum of disease progression, from an asymptomatic premalignant state, namely monoclonal gammopathy of undetermined significance (MGUS), to smoldering multiple myeloma (SMM), and finally to symptomatic MM, commonly with MM-associated end-organ damage, also known as CRAB (hyperCalcemia, Renal failure, Anemia, and Bone disease) [3,4].

Currently, MM is identified and staged according to the International Myeloma Working Group (IMWG) criteria. IMWG criteria outlines the MM diagnosis as clonal bone marrow plasma cells > 10%, or biopsy-proven bony or extramedullary plasmacytoma, in addition to one or more of the following: evidence of end organ damage (hypercalcemia, renal insufficiency, anemia and/or osteolytic bone lesions) and/or biomarkers positive for malignancy (serum-free light chain ratio, and clonal plasma cell proportion), and the presence of more than one focal marrow lesions on MRI [5]. There are several limitations to some of the tests involved in these criteria. The biomarkers outlined above are of limited value in non-secretory or extramedullary MM cases [3,4,5]. While bone marrow biopsy is the gold standard diagnostic test and is required for the assessment of treatment response, it is invasive, with a risk of local hemorrhage and infection, and it is not representative of the whole spectrum of the marrow [6].

Earlier in 2009, the consensus statement of the IMWG recommended conventional radiography (CR) as the standard imaging modality to stage MM, both in new cases and for relapsed patients, being widely available and inexpensive [5]. However, radiographs lack sensitivity and specificity, and they poorly detect extra-osseous lesions and diffuse medullary invasion. Therefore, skeletal surveys have been replaced with more sophisticated imaging techniques (computed tomography (CT), magnetic resonance imaging (MRI), and positron emission tomography (PET)) in many tertiary centers [7,8]. Newer staging systems incorporate more sensitive imaging modalities, such as whole-body MRI (WB-MRI) or ^18^F-fluorodeoxyglucose (FDG) PET/CT into the staging system [9]. The need for whole-body imaging in MM is related to the fact that myelomatous lesions can potentially affect any bony segment in the body.

Once it has been diagnosed and staged, as with most other malignancies, MM treatment management is also dependent on bone imaging to determine the extent of tumor cell burden, both for prognostic stratification and to assess post-therapy changes [10,11].

This paper describes available advanced imaging techniques in an assessment of MM, with a greater focus on whole-body diffusion weighted imaging (WB-DWI).

## 2. MRI

### 2.1. Conventional Whole-Body MRI (WB-MRI)

According to IMWG guidelines, MRI should be considered as the complementary adequate imaging modality for the diagnosis of MM, as it is more sensitive than FDG-PET/CT [12,13]. As opposed to computed tomography (CT) or positron emission tomography (PET), an advantage of MRI is that it causes no radiation exposure. In contrast to ^18^F-FDG PET/CT, MRI is widely available, relatively faster, has no pre-scan diet requirement, and is not reliant on the metabolic activity of tumor cells. This is specifically important in the case of MM, as frequent follow-up imaging is necessary, and survival time is increasing in light of recent therapeutic advancements. Conventional MRI can relatively accurately measure the size and extension of MM pathology, but it takes longer to interpret and requires more training to read than other imaging modalities [14,15,16,17].

Nearly half of MM lesions may be missed when imaging only covers the spine [14]. Thus, for a more sensitive assessment, MRI should cover the entire axial skeleton and the proximal appendicular skeleton, defining the idea of the “whole body” MRI technique. WB-MRI has gained popularity since the last decade. for diagnosing and assessing the treatment response in MM by providing morphological information on tumor spread. Around one-third to a half of MM patients show diffuse infiltration or focal deposits on WB-MRI [18,19].

The Myeloma Response Assessment and Diagnosis System (MY-RADS) comprehensively characterizes the myeloma state at diagnosis, initiation of treatment, and during follow-up, as the disease course changes in response to therapy. MY-RADS recommendations will help to improve response assessments by increasing standardization, and by decreasing the variations seen in the acquisition, interpretation, and reporting of whole-body MRI. For response assessments, the classified response assessment category (RAC) is according to anatomical regions. For each region, the RAC should use a five-point scale as follows: (1) highly likely to be responding; (2) likely to be responding; (3) stable; (4) likely to be progressing; and (5) highly likely to be progressing. MY-RADS functions to categorize patients with regard to specific disease patterns to aid in clinical trial stratification [20].

Different MRI sequences are being used based on imaging protocols of the radiology department [21,22,23,24]. Fat-suppressed T2-weighted (FS T2W), Short Tau Inversion recovery (STIR), and T1-weighted (T1W) MRI sequences are most commonly utilized for MM [23,24,25,26,27]. The signal intensity on FS T2W and STIR images correlates with the plasma cell concentration in the bone marrow. However, it lacks enough accuracy to differentiate between hyperplastic/red marrow and myeloma marrow. Changes on T1W images occur relatively late in the course of disease, and might differentiate MM from MGUS and SMM, although to a lesser extent compared to FS T2-WI, but they offer increased specificity [28]. While the T1W, STIR, and T2W sequences are more frequently used, diffusion weighted imaging (DWI) is the most promising MRI sequence for distinguishing MM from MGUS and SMM through visual assessment, and for the qualitative evaluation of lesion activity and treatment response [28].

### 2.2. Whole-Body Diffusion Weighted Imaging (WB-DWI)

DWI is a functional MRI sequence that can quantitatively evaluate tissue cellularity by measuring the random thermal movements of water molecules (Brownian motion) using the apparent diffusion coefficient (ADC) map [29]. Although DWI MRI was limited to brain imaging for many years due to its sensitivity to motion, its use has been recently expanded to include other anatomic locations, due to the introduction of echoplanar imaging and the use of fast acquisition techniques capable of capturing images during breath holds [30]. Thus, DWI MRI, including WB-DWI, have revolutionized the assessment of myeloma lesions from simple evaluations based on size to quantitative data based on free water molecule movement and tissue cellularity [29,31].

From 2010 on, the utilization of WB-DWI in MM has steadily increased [30,31,32,33]. WB-DWI can be performed relatively rapidly with low technical and operational efforts when added to the standard WB-MRI protocol. WB-DWI keeps the aforementioned advantages of WB-MRI, and adds further details to morphological imaging, as compared to the conventional sequences of WB-MRI during the assessment of treatment response and extramedullary disease, which are critical for MM management [21].

ADC mapping, which is derived from DW images, can distinguish bone marrow changes in active myeloma from those in remission, providing clinically relevant data on tumor viability [34]. ADC values above 600–700 μm^2^/s in a non-treated and newly diagnosed patient with multiple myeloma could be used to increase confidence for the diagnosis of diffuse marrow involvement, while normal marrow ADC value mostly falls below 600 μm^2^/s, with even lower values in elderly patients with prominent fatty marrow [32,35,36,37]. Koutoulidis et al. reported a higher ADC value for diffuse MM patterns in imaging comparing to focal lesions, and they found that an ADC value of >548 μmm^2^/s shows 100% sensitivity and 98% specificity for comparing a diffuse pattern of myeloma infiltration, than normal marrow [36]. Messiou et al. reported a significant decrease in the ADC values of MM patients who were responders from 4 weeks to 20 weeks after treatment, while patients with stable or progressive disease did not show a significant decrease in ADC value within the same time period [35].

If cellularity was the dominant factor in determining the ADC value in myeloma marrow, there should be a negative correlation between the ADC value and marrow cellularity. However, this is not the case. For this reason, cellularity is not the main factor affecting the DWI-ADC image signal in MM. Recent studies on liver fibrosis and pancreatic cancer have suggested that the perfusion effect on the measured ADC value dominates over hypercellularity [30,38,39]. This has been suggested for MM as well [40], as the ADC number decreases with hypercellularity and increases with hypervascularity, which parallels hypercellularity in the myeloma marrow [30]: the net effect is an increase of the ADC value in MM relative to the normal reference. This explains the Intravoxel Incoherent Motion (IVIM) MRI advantage for MM diagnosis/monitoring, which will be discussed in the following section. Employing lower b-values offer a better Signal to Noise Ratio (SNR), whereas higher b-values are more accurate for detecting MM lesions (Figure 1 and Figure 2) [41]. Most of the studies on DWI in MM have been conducted with a high b-value of 600–800, which offers a good MM lesion detection rate at a reasonable SNR [42,43,44].

One of the main advantages to WB-DWI is that it provides an excellent visual contrast between the normal marrow and bone marrow lesions, differentiating them with a higher sensitivity than the conventional MRI sequence, radiologic skeletal survey, or PET/CT scan [45,46,47]. For example, lesion conspicuity is greater in DWI as compared to the conventional T1-MRI and STIR sequences, and has a higher lesion detection rate compared to PET/CT [48,49].

DWI was able to identify 11% more patients than PET/CT in a cohort of 227 patients, and had a sensitivity of 77% versus 47% in PET/CT compared to a conventional MRI and CT in a smaller study of 24 patients [50]. WB-DWI also offers an excellent interobserver agreement for quantifying the disease burden in MM, both for the whole-body assessment and regional evaluation for any location across the body [48,51]. This superiority offers a substantial impact on treatment planning and patient classification, and it allows clinicians to make better decisions [35,52]. Importantly, WB-DWI has also been found to be useful in differentiating the stages of monoclonal gammopathies [35,52]. In addition, WB-DWI provides more reliable differentiation between benign tumors and pathological vertebral compression fractures [49]. DWI differentiates benign vertebral body collapse from malignant fractures with a reported sensitivity and specificity of 95.6% and 90%, respectively [36,48].

DWI-MRI is also useful in evaluating treatment efficacy, remission, and prognostication. ADC images, similar to other functional imaging modalities, account for the heterogeneous pathologic distribution and patchy marrow infiltration in MM (which is accentuated after relapse) [53], and thus can help to monitor the treatment response in MM patients.

With effective treatment, responders show an increased ADC value at 4–6 weeks post-treatment: hemorrhage, edema, vascular congestion, and liquefaction necrosis from tumor cell death contribute to an increased diffusivity [35,46]. ADC then decreases at 20 weeks after therapy, due to a reappearance of normal fatty marrow. Conversely, patients who are resistant to therapy show a persistent marrow hyperintensity on DWI, and hypointensity on ADC images [35,37,54,55].

DWI images have also been proven to detect prognostically relevant residual focal lesions with a higher sensitivity than PET and PET/CT [56,57]. For a diffuse pattern of myeloma infiltration, there are controversies regarding the correlation between the ADC signal and therapy response, as some studies support its use [47] while others do not [47,58]. A previous study has related this issue to the slower transition from replaced to recovered fatty marrow in focal MM lesions that make it possible to be captured when imaged post-treatment [36]. WB-MRI with DWI also offers a quantitative analysis of the entire bone marrow after treatment, which is invaluable for clinicians directing further need for therapies and remission.

The limitations of WB-DWI include some issues with a definitive identification of lesions, and with resolution. The ADC value of MM lesions increases early after effective treatment, and for an interpreter who is blind to previous exams, this might resemble a hemangioma. However, when previous images are not available, corresponding T1 and T2-weighted imaging solves this issue. Additionally, DWI suffers from some limitations in resolution, rendering, and field of view (FOV). Therefore, when interpreted with the available anatomical data, DWI is an important support for MM, and should be added to morphologic imaging sequences in WB protocols; and it currently cannot be relied on as a stand-alone imaging modality [49].

### 2.3. IVIM MRI

IVIM theory was first introduced by Le Bihan et al. more than two decades ago, describing the role of the perfusion effects on the significant signal decay at b-values of less than 300 s/mm^2^ [59]. The recorded ADC value is influenced by a combination of tissue microarchitecture/cellularity and perfusion. Given that the routine clinical implementation of DWI does not include a complete set of b-values (low, intermediate, and high), it does not provide information on the perfusion component of the signal derived from the imaged tissues [59].

As previously mentioned, in MM, the impact of the perfusion component on the ADC value seems to be larger than the effect of hypercellularity; hence, IVIM may be even more promising than DWI for MM patients [59]. IVIM decontaminates the ADC maps from the so-called “pseudo-diffusion” (D*), and marks out a part of the ADC signal that arises from any process other than diffusion, which is predominantly microperfusion at the capillary level. By taking three or more DWI images that are set at low, intermediate, and high b-values, two or more ADC maps could be generated, where ADC maps of low to moderate b-values reflect a combination of perfusion (D*: pseudo or enhanced diffusivity) and true diffusion (D). Subtracting these ADC maps will achieve tissue vascularity, which would be of added value to predict prognosis and assess treatment response in MM.

IVIM parameter D, the molecular diffusion coefficient, has been reported to be significantly higher in a diffuse pattern of myeloma marrow infiltration [60]. IVIM parameters are also associated with serum levels of biomarkers, showing that IVIM could be another useful tool for the prognostication of disease activity in MM [61].

### 2.4. Marrow Fat Quantification Techniques

Recently, gradient echo-based Dixon MRI as a fat quantification method approach has been evolved to include the generation of four separate image types: in-phase (IP), out-of-phase (OP), water-only (WO), and fat-only (FO) [62,63,64]. This method has been used for anatomical WB-MRI in MM, and it has several benefits over conventional T1- or T2-weighted imaging. The fat fraction is obtained using OP and IP images, and it can be efficient for both the lesion detection and the response assessment of bone lesions in MM patients [62]. It also has been shown that T2 Dixon fat-only Dixon images are more efficient by providing higher lesion detection rates compared to in-phase images alone in multiple myeloma [65].

### 2.5. Whole-Body Dynamic Contrast-Enhanced MRI (WB-DCE MRI)

WB-DCE MRI refers to the acquisition of serial images pre- and post-contrast administration, which can provide functional data on marrow infiltration in MM. WB-DCE MRI data can be computed into time-signal intensity curves providing a quantitative assessment of myeloma marrow infiltration, which could be valuable for diagnosis, prognosis, and monitoring treatment response. With regard to diagnosis, higher peaks and steeper slopes of WB-DCE MRI curves are associated with a higher percentage of plasma cell infiltration [66,67,68]. Prior studies have also found WB-DCE MRI curves to be well-correlated with MM disease activity and serum biomarkers [67]. In terms of prognosis, the relative signal enhancement of marrow in SMM predicts its rapid progression into symptomatic disease, and more severe enhancement in progressive MM predicting a shorter progression-free survival [69,70]. After treatment, persistent abnormally elevated peaks of marrow enhancement and the foci of early enhancement portend disease progression or relapse (poor treatment response) [71]. Due to the paucity of research on standard acquisition protocols and reporting systems, WB-DCE MRI is not presently recommended for daily practice [72].

## 3. ^18^F-Fluorodeoxyglucose Positron Emission Tomography (18FDG PET)

### 3.1. ^18^FDG PET

^18^F-fluorodeoxyglucose (FDG) is the most commonly used radiotracer in PET imaging for MM [73]. FDG uptake, which is widely semi-quantitatively assessed via the maximum standardized uptake value (SUVmax), is known to be correlated with biological tumor aggressiveness [74]. ^18^FDG PET may over- or underestimate MM disease activity; for example, over-calling pathologic fractures or under-identifying small sized lesions. Moreover, SUVmax measures ^18^FDG uptake in a single-voxel region of interest, which might emit a distorted signal due to noise, reconstruction protocols, and post-processing (if applicable). Additionally, a single-voxel measurement may not be representative of marrow disease in its entirety [75]. Reporting SUVmean and SUVpeak instead of SUVmax addresses these setbacks, but is rarely performed in clinical practice [76]. Additionally, recent studies have suggested that measuring total lesion glycolysis (TLG) and metabolic tumor volume (MTV) may represent the tumor burden/activity and the patient’s prognosis more accurately [77,78]. However, these measurements can be time-consuming and not routinely performed in clinical practice. Hybrid imaging and adding the morphological data from CT or MRI into the functional data from PET imaging resolves these issues to a considerable extent.

### 3.2. ^18^FDG PET/CT

^18^FDG PET/CT is a sensitive diagnostic modality for both medullary and extramedullary plasma cell dyscrasias, and can accurately detect MM lesions, assess treatment response, and predict prognosis and progression in MM [77,79,80,81]. The wider employment of ^18^FDG PET/CT has also resulted in a significantly increased detection rate for extramedullary MM at diagnosis compared to older imaging techniques [82].

The International Myeloma Working Group (IMWG) recently introduced the evaluation of minimal residual disease (MRD) within the multiple myeloma (MM) response criteria. Currently the most powerful predictor of favorable outcomes over long-term follow-up, MRD negativity can be assessed both inside and outside of the bone marrow. Functional imaging techniques such as PET/CT and magnetic resonance imaging (MRI) serve in sensitive response assessment, and have been shown to be promising in terms of evaluating the response to treatment. Not only have they helped to assess MRD status in MM patients, but they also provide a global representation of the tumor burden by including several prognostic markers in addition to lesion type [83].

The sensitivity of ^18^FDG PET/CT for focal MM lesions has been reported to be more than WB-MRI, roughly estimated at 75% [84]; however, a diffuse pattern of marrow infiltration is better detected using WB-MRI. Additionally, ^18^FDG PET/CT was shown to have more promising results in having a higher impact on clinical decisions than WB-MRI in MM patients in terms of prognosis and management [85]. In a recent meta-analysis, a high ^18^FDG uptake significantly and independently predicted a shorter overall and progression-free survival in MM patients [81] when considering whole-body burden disease [81].

WB-MRI with DWI provides a non-invasive and quantitative assessment of the entire bone marrow after treatment. Based on a study by Torkian et al., DWI had a pooled sensitivity of 78% (95% CI: 72–83) and a specificity of 73% (95% CI: 61–83) in distinguishing responders from non-responders, emphasizing the prominent role of DWI for treatment response assessment in patients with MM [86]. In a cohort of 49 cases, WB DWI has been shown to be more sensitive than ^18^FDG PET/CT for detecting intramedullary lesions in all regions except the skull, both in patients with a new diagnosis and previously treated patients. Additionally, WB DWI has been shown to have a sensitivity equivalent to that of ^18^FDG PET/CT for detecting extramedullary lesions [87].

### 3.3. ^18^FDG PET/MRI

Finally, hybrid ^18^FDG PET/MRI techniques combine the morphological information provided by MRI and the metabolic and functional data furnished via ^18^FDG PET imaging, which allows for both the ability to detect the marrow foci of myeloma infiltration and to assess prognosis and treatment response. ^18^FDG PET/MRI increases the visibility of focal MM lesions at diagnosis and initial staging, and localizes residual disease activity after treatment [12]. ^18^FDG PET/MRI has been reported to have a higher lesion detectability rate than ^18^FDG PET/CT in evaluating skeletal lesions [88]. However, Sachpekidis et al. concluded that these two techniques are equally sensitive in detecting MM lesions [89]. As with WB-DCE MRI, there are few studies on a standard acquisition protocol and reporting system in ^18^FDG PET/MRI for MM; thus, further studies are warranted to test repeatability and validity before ^18^FDG PET/MRI can be considered as a viable tool in a routine imaging work-up of MM.

## 4. Conclusions

In the last two decades, novel imaging techniques have been developed or employed for MM to provide more information that can guide not only diagnosis and staging, but also treatment efficacy. These imaging techniques, due to their low invasiveness and high reliability, have gained significant clinical attention and have changed the clinical practice. The development of functional MRI sequences such as diffusion weighted imaging (DWI) and Intravoxel Incoherent Motion (IVIM) has made the functional assessment of lesions feasible.

DWI-MRI provides a high utility tool for not only the diagnosis and initial staging of plasma cell dyscrasias such as MM, but also for the evaluation of treatment efficacy and for re-staging. It can reliably detect tissue hypercellularity with high sensitivity, challenging the use of other procedures or imaging modalities for the management of MM. Therefore, DWI-MRI presents a promising option for clinicians engaged in the management of plasma cell dyscrasias like MM.

## Figures and Tables

**Figure 1 diagnostics-12-02182-f001:**
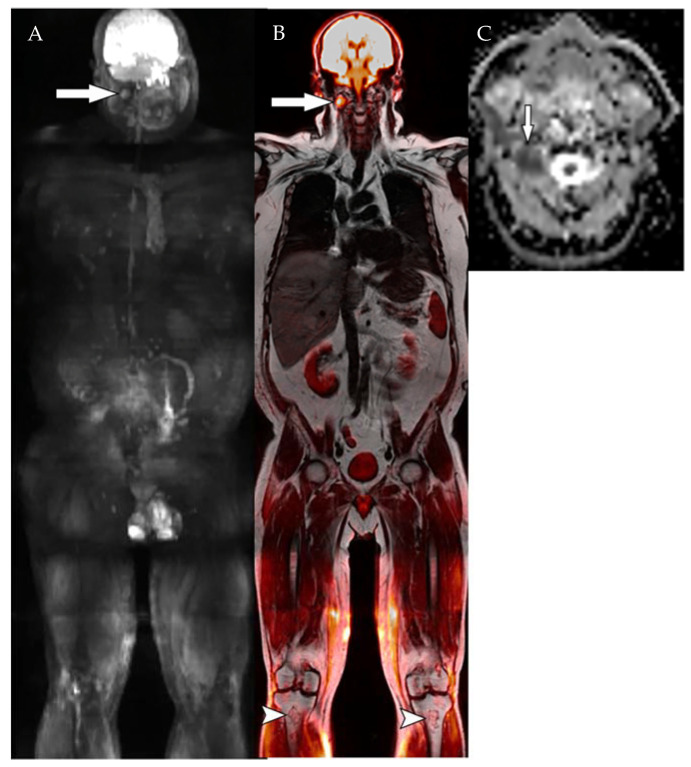
A 45-year-old man with hyposecretory IgG kappa multiple myeloma (MM) and a history of autologous stem cell transplantation 3 years before imaging. Patient had received radiation therapy to L1 vertebral body 2 years ago, and has been on maintenance therapy (carfilzomob, pomalidomide, and dexamethasone) for 6 months. WB-MRI images failed to demonstrate any osseous lesion, but showed a 12 mm extramedullary lesion deep to right parotid gland (arrow) on coronal high b-value (b = 800) DWI MIP image (**A**). Fused coronal DWI MIP and T1-weighted Dixon image (**B**) confirms the finding (arrow). Axial ADC map (**C**) confirms diffusion restriction (mean value of 0.8 (×10^−3^ mm^2^/s). Note areas of marrow infarction at bilateral proximal tibial metaphysis (arrowheads, (**B**)).

**Figure 2 diagnostics-12-02182-f002:**
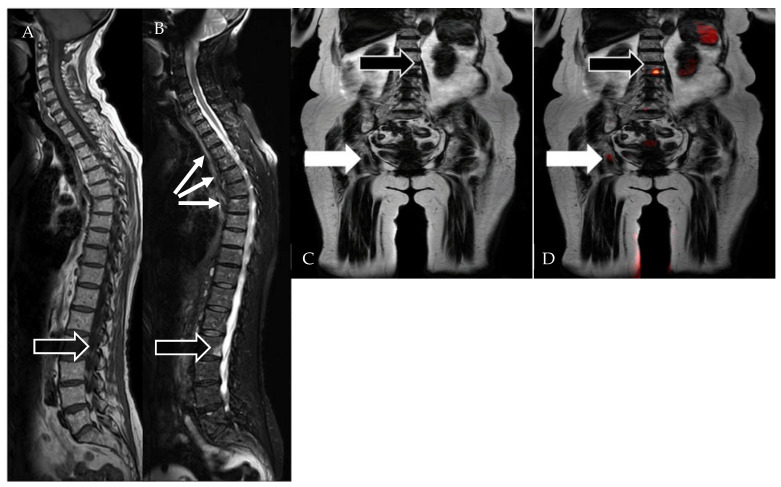
WB-MRI in a 75-year-old woman with mixed active and treated disease. (**A**) Sagittal T1-WI shows a lesion (black arrow) in the posterior and inferior L2 vertebral body. (**B**) Sagittal STIR image shows the L2 lesion (black arrow), as well as at several other sites of disease (thin white arrows) that were not obvious on the T1-WI. (**C**) Coronal T1 Dixon Fat image shows the L2 lesion (black arrow) and another lesion in the right ischium (white arrow). (**D**) Fused b800-Dixon image shows increased signal at L2 (black arrow) and to a lesser extent at the ischial lesion (white arrow).

## Data Availability

Not applicable.

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
