# Peer review of "Advanced Imaging in Multiple Myeloma: New Frontiers for MRI"

_diagnostics, 2022, doi:10.3390/diagnostics12092182_

Round 1

Reviewer 1 Report

The paper is focused on the role of new functional MRI techniques in the management of MM patients. However, when they were compared with other more traditional imaging techniques such as PET/CT their superiority does not emerge. Therefore, the Authors should make an effort to better analyze and discuss the role of different imaging techniques.

Author Response

08.14.2022

Andreas Kjaer, MD

Editor-in-Chief, Diagnostics

Diagnostics-1803647: " Advanced Imaging in Multiple Myeloma: New Frontiers for MRI”

Dear Dr. Kjaer,

Thank you very much for your prompt review and constructive comments on our manuscript (Diagnostics-1803647) entitled " Advanced Imaging in Multiple Myeloma: New Frontiers for MRI ". We are incredibly grateful to have the opportunity to submit a revised version of our manuscript. We have prepared the revision and implemented changes based on the provided comments.

In this response document, we have structured the responses in “Author Response” and have also cross-referenced numbered reviewer comments in the revised manuscript text.

In addition, we have uploaded a clean version of manuscript and an annotated version with all changes visible and cross-references to Reviewers’ comments. We hope that Diagnostics Editor and Reviewers will find our point-by-point responses to the comments satisfactory and agree with us that the comments and edits have substantially increased the quality of our manuscript.

Best wishes,

The Authors

 Reviewer 1

Comment 1: The paper is focused on the role of new functional MRI techniques in the management of MM patients. However, when they were compared with other more traditional imaging techniques such as PET/CT their superiority does not emerge. Therefore, the Authors should make an effort to better analyze and discuss the role of different imaging techniques.

Authors Response: Thank you for your insightful comments, we agree with your suggestion, but in this review paper we tried to have an organized framework to evaluate the benefits and challenges of novel and more functional imaging techniques used for management of patients with plasma cell dyscrasias, especially MM. We acknowledge that this is an important comment; however, the relevant published literature is heterogeneous. The development of functional MRI sequences such as Diffusion-Weighted Imaging (DWI) or Intravoxel Incoherent Motion (IVIM) has made functional assessment of lesions feasible. Also, in a recent systematic review project published by our group, the diagnostic performance of DWI for disease detection, staging of MM, and assessing response to treatment in MM patients were evaluated. It has been shown that DWI is not only a promising tool for the diagnosis of MM, but it is also useful in the initial staging and re-staging of the disease and treatment response assessment. This can aid clinicians with timely and optimal decision making on treatment strategy, which could be of prognostic significance.

The following part has been added in the last paragraph of “3.2. 18FDG PET/CT”:

WB-MRI with DWI provides a non-invasive and quantitative assessment of the entire bone marrow after treatment. Based on study by Torkian et al., DWI had a pooled sensitivity of 78% (95% CI: 72–83) and specificity of 73% (95% CI: 61–83) in distinguishing responders from non-responders, emphasizing the prominent role of DWI for treatment response assessment in patients with MM.

In a cohort of 49 cases, WB DWI has been shown to be more sensitive than PET/CT for detecting intramedullary lesions in all regions except the skull, both in patients with a new diagnosis and previously treated patients. Also, WB DWI has been shown to have the sensitivity equivalent to that of PET/CT for detecting extramedullary lesions.

Reviewer 2 Report

1. Introduction

Page 1, lines 43-44 Correct the myeloma defining events (delete “serum and urine M protein concentrations” and add “presence of more than one focal lesion on MRI”)

2. MRI

Expand on the role of ADC values (report range of normal and abnormal values) for diagnosis of focal lesions and diffuse involvement as well as for treatment response assessment (use references 50, 49, 54 and Messiou et al Radiology 2019; 291:5–13, Koutoulidis et al Br J Radiol 2018; 91:20170389).  

Page 2, lines 79-80 “Up to one third of cases can upstage the disease from Solitary Bone Plasmocyotma (SBP) 79 to MM using MRI.” Add reference for this statement and correct typo (plasmacytoma).

Page 2, line 92 Define DWI

Page 3, line 108 Delete “at a”

Page 3, line 111-112 Delete “lymph node involvement”

Page 5, line 148 Change “bone lesions” to “bone marrow lesions”

Expand on the role of MRI and WB-MRI for treatment response asssessment in MM. Mention the MY-RADS criteria and other relevant references.

3. 18F-Fluorodeoxyglucose Positron Emission Tomography (18FDG PET) 

Page 7, lines 237-239 Delete the part of the sentence after “daily practice”

In the MRI section (page 5 lines 147-151) you state: One of the main advantages to WB-DWI is that it provides an excellent visual contrast between normal marrow and bone lesions, differentiating them with a higher sensitivity than conventional MRI sequence, radiologic skeletal survey, or PET/CT scan (41-43). For example, lesion conspicuity is greater in DWI as compared to conventional T1-MRI and STIR sequences and has a higher lesion detection rate compared to PET/CT (44, 45). 

whereas in the PET/CT section you write: The sensitivity of 263 18FDG PET/CT for focal MM lesions has been reported to be more than WB-MRI, roughly 264 estimated at 75% (82).

So, which is it? Is MRI more sensitive or less sensitive than PET/CT for the detection of focal MM lesions? 

Expand on the role of PET/CT for the definition of minimal residual disease (MRD) with appropriate references.

Figures 

Add units of measurement to ADC values in figure legends.

Where are the thin white arrows in Figure 2?

Author Response

08.14.2022

Andreas Kjaer, MD

Editor-in-Chief, Diagnostics

Diagnostics-1803647: " Advanced Imaging in Multiple Myeloma: New Frontiers for MRI”

Dear Dr. Kjaer,

Thank you very much for your prompt review and constructive comments on our manuscript (Diagnostics-1803647) entitled " Advanced Imaging in Multiple Myeloma: New Frontiers for MRI ". We are incredibly grateful to have the opportunity to submit a revised version of our manuscript. We have prepared the revision and implemented changes based on the provided comments.

In this response document, we have structured the responses in “Author Response” and have also cross-referenced numbered reviewer comments in the revised manuscript text.

In addition, we have uploaded a clean version of manuscript and an annotated version with all changes visible and cross-references to Reviewers’ comments. We hope that Diagnostics Editor and Reviewers will find our point-by-point responses to the comments satisfactory and agree with us that the comments and edits have substantially increased the quality of our manuscript.

Best wishes,

The Authors

Reviewer 2

Comment 1: Introduction, Page 1, lines 43-44 Correct the myeloma defining events (delete “serum and urine M protein concentrations” and add “presence of more than one focal lesion on MRI”)

Authors Response: Many thanks for this insightful comment. We applied this change in the updated version.

Comment 2:  MRI, Expand on the role of ADC values (report range of normal and abnormal values) for diagnosis of focal lesions and diffuse involvement as well as for treatment response assessment (use references 50, 49, 54 and Messiou et al Radiology 2019; 291:5–13, Koutoulidis et al Br J Radiol 2018; 91:20170389).  

Authors Response: Thanks for bringing that to our attention, we added this section in our updated version:

ADC values above 600–700 μm2/ sec in a nontreated and newly diagnosed patient with multiple myeloma could be used to increase confidence for the diagnosis of diffuse marrow involvement, while normal marrow ADC value mostly falls below 600 μm2/s with even lower values in elderly patients with prominent fatty marrow.Koutoulidis et al. reported a higher ADC value for diffuse MM pattern in imaging comparing to focal lesions, and they found that ADC value of >548 μmm2/sec shows  100% sensitivity and 98% specificity for comparing diffuse pattern of myeloma infiltration than normal marrow. Messiou et al. reported a significant decrease in ADC values of MM patients who where responders from 4 weeks to 20 weeks after treatment, while patients with stable or progressive disease did not showed a significant decrease in ADC value in same time period.

Comment 3: Page 2, lines 79-80 “Up to one third of cases can upstage the disease from Solitary Bone Plasmocyotma (SBP) 79 to MM using MRI.” Add reference for this statement and correct typo (plasmacytoma).

Authors Response: Thanks for bringing that to our attention. That section has updated and corrected.

Comment 4: Page 2, line 92 Define DWI

Authors Response: Many thanks for this insightful comment. We applied this change in the updated version.

Comment 5: Page 3, line 108 Delete “at a”

Authors Response: Thanks for bringing that to our attention. We applied this change in the updated version.

Comment 6: Page 3, line 111-112 Delete “lymph node involvement”

Authors Response: Thanks for bringing that to our attention. We applied this change in the updated version.

Comment 7: Page 5, line 148 Change “bone lesions” to “bone marrow lesions”

Authors Response: Thanks for bringing that to our attention. We applied this change in the updated version.

Comment 8: Expand on the role of MRI and WB-MRI for treatment response assessment in MM. Mention the MY-RADS criteria and other relevant references.

Authors Response: Thanks for bringing that to our attention, we added this section in our updated version:

The Myeloma Response Assessment and Diagnosis System (MY-RADS) comprehensively characterizes the myeloma state at diagnosis, initiation of treatment, and during follow-up as the disease course changes in response to therapy. MY-RADS recommendations will help to improve response assessments by increasing standardization and by decreasing the variations seen in the acquisition, interpretation, and reporting of whole-body MRI. For response assessments, the classified response assessment category (RAC) is according to anatomic regions. For each region, the RAC should use a five-point scale as follows: 1, highly likely to be responding; 2, likely to be responding; 3, stable; 4, likely to be progressing; 5, highly likely to be progressing. MY-RADS functions to categorize patients with regard to specific disease patterns to aid in clinical trial stratification.

Comment 9: 18F-Fluorodeoxyglucose Positron Emission Tomography (18FDG PET) , Page 7, lines 237-239 Delete the part of the sentence after “daily practice”

Authors Response: Thanks for bringing that to our attention. We applied this change in the updated version.

Comment 10: In the MRI section (page 5 lines 147-151) you state: One of the main advantages to WB-DWI is that it provides an excellent visual contrast between normal marrow and bone lesions, differentiating them with a higher sensitivity than conventional MRI sequence, radiologic skeletal survey, or PET/CT scan (41-43). For example, lesion conspicuity is greater in DWI as compared to conventional T1-MRI and STIR sequences and has a higher lesion detection rate compared to PET/CT (44, 45). 

whereas in the PET/CT section you write: The sensitivity of 263 18FDG PET/CT for focal MM lesions has been reported to be more than WB-MRI, roughly 264 estimated at 75% (82).

 So, which is it? Is MRI more sensitive or less sensitive than PET/CT for the detection of focal MM lesions? 

Authors Response: Thanks for your comment. Based on the current literature, In terms of sensitivity to detect focal MM lesions: WB-DWI is more sensitive than PET/CT and PET/CT is more sensitive than conventional WB-MRI. We believe both statements are correct and changes have been made to clarify them.

Comment 11: Expand on the role of PET/CT for the definition of minimal residual disease (MRD) with appropriate references.

Authors Response: Thanks for bringing that to our attention, we added this section in our updated version:

The International Myeloma Working Group (IMWG) recently introduced the evaluation of minimal residual disease (MRD) within the multiple myeloma (MM) response criteria. Currently the most powerful predictor of favorable outcomes over long-term follow-up, MRD negativity can be assessed both inside and outside the bone marrow.

Functional imagine techniques like PET/CT and magnetic resonance imaging (MRI) serve in sensitive response assessment and have shown to be promising in terms of evaluating the response to treatment. Not only have they helped assess MRD status in MM patients, but they also provide a global representation of the tumor burden by including several prognostic markers in addition to lesion type.

Comment 12: Add units of measurement to ADC values in figure legends.

Authors Response: Thanks for your comment. The unit added in figure legends (x10-3mm2/s).

Comment 13: Where are the thin white arrows in Figure 2?

Authors Response: Thanks for your comment. The arrows added in figure legends.

Round 2

Reviewer 1 Report

The paper in the revised form is clearer and acceptable for publication